# Carcass and Meat Quality Traits of Medium-Growing Broiler Chickens Fed Soybean or Pea Bean and Raised under Semi-Intensive Conditions

**DOI:** 10.3390/ani12202849

**Published:** 2022-10-20

**Authors:** Siria Tavaniello, Antonella Fatica, Marisa Palazzo, Sanije Zejnelhoxha, Mengjun Wu, Luigi De Marco, Elisabetta Salimei, Giuseppe Maiorano

**Affiliations:** Department of Agricultural, Environmental and Food Sciences, University of Molise, 86100 Campobasso, Italy

**Keywords:** broiler chickens, diet, protein sources, genotype, meat quality, fatty acids

## Abstract

**Simple Summary:**

Soybean is an excellent protein source for many livestock species. However, due to the economic and environmental sustainability concerns, alternatives that can serve as full or partial replacements of soybean products are increasingly being explored. This study investigated the effects of the total replacement of flaked soybean with raw pea beans on carcass characteristics and meat quality of two medium-growing broiler chicken strains reared under semi-intensive conditions. Diet did not affect final body weight, carcass traits or meat quality. The fatty acid profile was significantly influenced by diet. Genotype had strong effects on slaughter body weight and carcass characteristics and slightly affected the fatty acid profile of meat. Further studies are needed to find the better inclusion rate in order to promote the use of peas in broiler diets, in other words, to find the right compromise between environmental and economic sustainability of the diet and promotion of the nutritional quality of the meat.

**Abstract:**

A study was carried out to evaluate the effect of the total replacement of flaked soybean (*Glycine max* L., SOY) with raw pea (*Pisum sativum L.*, PEA) on the carcass and meat quality traits of two medium-growing broiler strains (Kabir Rosso Plus, KB; New Red, NR). Birds were housed in 20 pens (five replications/groups, six birds each). At 83 days of age, 40 birds (2/replication) were slaughtered and the pectoral muscle (PM) was removed for analyses. Diet did not affect slaughter weight, carcass traits and meat quality. A pea diet determined a significant increase of MUFA and a decrease of PUFA, n-3 and n-6 PUFA; hence, the pea-fed group had a lower PUFA/SFA and a higher n-6/n-3 ratios compared to the soy-fed. NR chickens were heavier, with higher carcass and cut weights (*p* < 0.01) compared to KB chickens. Interactions (*p* < 0.05) between factors were found for PM weight and yield. Meat from NR had a higher (*p* < 0.05) pH. Fatty acids were slightly affected by genotype. Replacing soybean with pea adversely affects meat fatty acid composition in terms of nutritional profile.

## 1. Introduction

Consumers perceive the safety and quality of meat products as essential characteristics and are willing to pay more for naturally obtained meat products, paying particular attention to animal well-being. Poultry meat can be considered one of the most important sources of the cheapest and highest quality protein. However, poultry meat quality results from complex interactions between the genotype, age and sex of birds and their management system [1]. As well-known, diet is one of the key factors affecting the quantitative and qualitative characteristics of poultry meat and the profitability of production. Nowadays, soybean is the dominant source of protein available for inclusion in broiler diets. Chicken-meat production absorbs a considerable proportion of soybean meal (44% and 32% of soybean meal fed to animals in the USA and Europe, respectively; [2]). Considering that global world chicken meat production is projected to double by 2050, soybean meal consumption is expected to increase to 181 million tons in 2050 (reviewed by Selle et al. [3]). In most countries, imported soybean meal (derived from the oil extraction process) is mainly produced from genetically modified plants [4]. Soybean is a crucial feed ingredient for many reasons: the unsustainability of the utilization of soybean products for feeding animals instead of feeding humans; the increased soybeans price (e.g., in Italy: August 2022 + 40.3% compared to 2021 [5]); the prospect that after January 1, 2024, genetically modified materials may be banned in the EU, which will represent a serious problem in poultry production [6]. Consequently, the need to seek alternative protein ingredients, which are locally available and affordable, is urgently required. As previous studies indicate, the interest in using grain legumes such as peas (*Pisum sativum* L.), faba beans (*Vicia faba* L.) and lupins (*Lupinus* spp.) as alternatives to conventional protein sources (e.g., soybean products) in livestock diets is increasing [7,8,9]. Moreover, the mature seeds of most legume crops contain important carbohydrate components, including starches and sugars, and fat [10,11]. Current varieties of legumes contain low levels of antinutrients making them candidates for use as high-protein sources in poultry diet [8]. On the other hand, the results of our previous work [12] indicated that the replacement of soybean with raw pea beans could be an economically and environmentally sustainable feeding strategy for inner areas of central South Italy. Pea cultivation in the inner areas of the southern Apennines is practiced to alleviate the problems of cereal monoculture, as well as to improve the productive performance of mixed agro-forestry systems and connect these with livestock farming in an eco-sustainable farming perspective [13]. In addition, unlike other legumes, the pea is a flexible crop, fitting into many crop rotations due to its potential winter sowing and rapid seed ripening, which allows them to escape summer drought and high temperatures [14,15]. All these aspects may represent valid motivations for the sustainable use of peas in the inner Mediterranean areas. Nowadays, chicken-meat production mainly utilizes fast-growing genotypes housed indoors under climate-controlled conditions and fed balanced diets that guarantee high growth performance and feed efficiency. However, this type of production has a negative impact on traits such as well-being and disease resistance [16]. Therefore, throughout the world there is an increasing interest in alternative and less intensive poultry production systems that meet higher standards for animal welfare, for which medium- and slow-growing broiler strains have been recommended [17]. Indeed, several authors reported that medium- and slow-growing broiler strains perform better when referred to health issues than conventional strains [18], including more resilience to heat stress [19] and a meat more appropriate for a specialty or gourmet market [20,21,22]. The impact of the use of soybean meal and legume seeds in feed on the meat quality of fast-growing broiler chickens has been studied by other authors [8,23,24,25]. However, limited information is available on the effects of legume seeds on the carcass traits and meat quality of medium-growing chickens reared in a semi-intensive system. Therefore, the aim of the present study was to evaluate the effects of two dietary protein sources (soybean vs. pea bean) on the carcass traits and meat quality of two chicken genotypes (Kabir Rosso Plus vs. New Red) raised under traditional semi-intensive conditions.

## 2. Materials and Methods

### 2.1. Ethics

This on-field experiment was carried out as a routine production cycle in small-scale farming conditions. Research protocol was in accordance with the European Commission guidelines (2010/63/EU) concerning the protection of animals used for experimental and other scientific purposes.

### 2.2. Animal Trial

This trial was conducted in a private poultry farm located in Campania region (South Italy) in a hilly area at about 300 m above sea level (Benevento municipality). The experiment was conducted from October to December 2019, under semi-intensive rearing conditions. Twenty day-old medium-growing male chickens, Kabir Rosso Plus (KB, n = 60) and New Red (NR, n = 60), vaccinated according to the current commercial practice, were housed and randomly divided into 4 groups according to genotype (KB and NR) and dietary treatment (soybean, SOY and pea bean, PEA) of 5 replicate pens/group with 6 birds/pen (KB-SOY, NR-SOY, KB-PEA, NR-PEA). Birds were reared in similar spatial and environmental conditions as reported by Fatica et al. [12]. The feeding trial lasted 36 days (from 47 to 83 days of age) preceded by an adaptation period as reported by Fatica et al. [12]. Two isonitrogenous and isoenergetic experimental diets were formulated according to birds’ nutritional needs, including either flaked soybean (SOY) or pea bean (PEA); moreover, fava bean (*Vicia faba* L.) and wheat products, locally available, were included in both diets. Ingredients, and the chemical and fatty acid composition of the diets are reported in Table 1.

### 2.3. Slaughter Surveys and Physico-Chemical Analyses

At 83 days of age, a total of 40 randomly chosen birds (10 birds from each experimental group, 2 birds for each replication) were individually weighed, labelled and processed under commercial conditions. The carcass weight was obtained by removing the head, neck, shanks, and abdominal fat from bled, plucked, and eviscerated chickens; the carcass yield was calculated. Then, breast (including pectoralis major and pectoralis minor muscles), leg (thigh + drumstick), and wing weights were recorded; cut yields were calculated based on hot carcass weight. On the right pectoral muscle (PM), pH and color were recorded at 24 h post-mortem. The pH was measured on the upper part of the left-side breast fillet using a portable pH meter (FiveGo, Mettler-Toledo, Switzerland) equipped with a penetrating glass electrode. Tri-stimulus color coordinates (lightness, L***; redness, a***; yellowness, b***) were measured on the bone-side surface of the left-side breast fillet using a Chroma Meter CR-300 (Konica Minolta B. S. Italia Spa, Milan, Italy). The left PM was vacuum packaged and frozen (−20 °C) until chemical analysis. Water-holding capacity, expressed as expressible juice, was measured on PM 24 h after chilling using the press method [26].

### 2.4. Total Lipids and Fatty Acid Profile

Lipid extraction from breast muscle was performed according to the chloroform: methanol extraction procedure [27]. Fatty acids (FA) were quantified as methyl esters (FAME) using a gas chromatograph GC Trace 2000 (ThermoQuest EC Instruments, Milan, Italy) equipped with a flame ionization detector (260 °C) and a fused silica capillary column (Zebron ZB-88, Phenomenex, Torrance, CA, USA) 100 m × 0.25 mm × 0.20 μm film thickness. Helium was used as a carrier gas. The column temperature was held at 100 °C for 5 min, then raised 4 °C/min up to 240 °C and maintained for 20 min. The individual fatty acid peaks were identified by comparison of retention times with those of known mixtures of standard fatty acids (37 Component FAME MIX and docosapentaenoic acid (cis-7,10,13,16,19), Supelco, Bellofonte, PA, USA) run under the same operating conditions. Results are expressed as percentages of the total FA identified. To assess the nutritional implications, the ratio of n-6 PUFA to n-3 PUFA (n-6/n-3) and the ratio of polyunsaturated fatty acids (PUFA) to saturated fatty acids (SFA) (P/S) were calculated. Moreover, to evaluate the risk of atherosclerosis and the potential aggregation of blood platelets, the atherogenic index (AI) and the thrombogenic index (TI) were calculated according to the formulas suggested by Ulbricht and Southgate [28].

### 2.5. Cholesterol Content

Cholesterol was extracted using the method of Maraschiello et al. [29] and then quantified by HPLC. A Kontron HPLC (Kontron Instruments, Milan, Italy) model 535, equipped with a Kinetex 5μ C18 reverse-phase column (150 cm × 4.6 mm × 5 μm; Phenomenex, Torrance, CA), was used. The HPLC mobile phase consisted of acetonitrile:2-propanol (55:45, vol./vol.) at a flow rate of 1.0 mL/min. The detection wavelength was 210 nm. The quantitation of muscle cholesterol content was based on the external standard method using a pure cholesterol standard (Sigma, St. Louis, MO, USA).

### 2.6. Statistical Analyses

Statistical analyses of the data were performed using IBM SPSS Statistic Data Editor version 25 (Chicago, IL, USA). Data were analyzed by a factorial ANOVA including dietary treatment, genotype, and their interaction. Each bird was considered as the experimental unit. Data are expressed as mean ± standard error of the mean (SEM) and differences were considered significant for *p* < 0.05.

## 3. Results and Discussion

### 3.1. Carcass and Physico-Chemical Traits

Growth performance results (body weight, feed intake, and feed conversion ratio) regarding all reared chickens (120) are reported by Fatica et al. [12]. Final body weight (BW) and carcass traits of slaughtered chickens are reported in Table 2.

Diet did not affect (*p* > 0.05) final BW and carcass traits. Similarly, Laudadio and Tufarelli [30] found that the different dietary protein sources (wheat middlings-based diet containing soybean vs. micronized–dehulled peas) did not affect the BW of Hubbard broiler chickens. Our carcass traits results are consistent with those of previous studies regarding the effects of diets containing peas or soybean [23,30]. Dal Bosco et al. [31] found a lighter carcass in slow-growing chickens fed with the partial substitution of soybean with extruded fava bean, while no differences were reported for dressing percentage and other carcass traits. Differently, Biesek et al. [8] found a lower body weight gain and carcass cuts (breast and legs) weight in chickens (Ross 308) fed with legumes (pea or fava bean) compared to those fed with genetically modified soybean meal; however, they did not observe significant differences in neck and wing weights, and in carcass, breast, neck, and wing yields.

Genotype had strong effects on the slaughter BW and carcass characteristics (Table 2). Compared with KB chickens, NR chickens were heavier (*p* < 0.01) with higher carcass and cuts (breast, legs and wings) weights (*p* < 0.01); in addition, NR showed higher carcass and breast yields (*p* < 0.01), while leg and wing yields were similar (*p* > 0.05) between the studied genotypes. The results observed for the carcass traits could be explained by the greater adaptability of NR birds to the management and environmental farm conditions as reported in our study [12] and by the positive correlations between live weight and the weights of carcass parts [32]. Breast muscle, the main valuable part of the carcass, expressed as a percentage (ranging from 21.18 to 22.67 %), calculated as breast meat weight divided by hot carcass weight, is similar to that reported in the medium-growing chicken Hubbard JA757 [33] and higher compared with slow-growing ISA Dual chickens. Significant interaction (*p* < 0.05) effects of diet and genotype on the breast weight and yield were observed (Table 2). This result highlights that the KB birds had a higher breast weight and breast yield when fed with a soy diet, while NR birds showed higher values when fed with a pea diet (breast weight: KB-SOY: 348.2 g; NR-SOY: 396.6 g; KB-PEA: 307.6 g; breast yield: NR-PEA: 424.4 g; KB-SOY 21.63%; NR-SOY: 21.93% g; KB-PEA: 20.72%; NR-PEA: 23.41%).

Physico-chemical properties of breast muscle are reported in Table 3.

The use of pea beans did not significantly affect meat pH, color and WHC. Similarly, other studies [8,23,30] on the use of pea as a substitute of soybean meal did not show differences in meat quality traits (pH and color). Differently, Dal Bosco et al. [31] reported that a fava bean diet increased the pH values of breast muscle as compared to a soybean diet, but it did not influence meat color. Laudadio and Tufarelli [30] observed higher WHC values in the meat of birds fed with peas in comparison with birds fed the soybean diet.

Genotype influenced only the pH of breast meat: NR birds showed a higher (*p* < 0.05) ultimate pH value than the KB birds (Table 3). The ultimate pH is known to affect the structure of myofibrils and consequently the WHC and the color of the meat [34]; this relationship is not confirmed in our study. However, pH values found in the present study were in normal ranges [35] and do not indicate any problems. Differences in ultimate pH were generally found between slow-growing chickens and fast–medium-growing ones [33,36,37].

### 3.2. Total Lipid, Cholesterol and Fatty Acid Composition

The replacement of soybean with pea beans in the diet did not affect (*p* > 0.05) lipid and cholesterol contents in chicken breast muscles (Table 4). Likewise, Dotas et al. [23] did not find any significant effect of diet with an increasing inclusion of field pea on total lipid content in the breast and drumstick muscles of Ross 308 broiler chickens. A similar result was reported by Kuźniacka et al. [24] with fava bean used as a substitute of soybean meal in the diet.

Genotype did not affect lipid and cholesterol contents in breast muscle (*p* > 0.05). Chodová et al. [33] observed a higher amount of lipids in fast- (Ross 308) than in medium- (Hubbard JA757) and slow-growing (ISA Dual) chickens; however, the differences between Hubbard JA757 and ISA Dual genotypes were low. On the other hand, no difference was observed for the cholesterol content between the genotypes studied by Chodová et al. [33]. Cholesterol content has become an important component in composition studies on meat and poultry products determining one of the attributes of the perception of quality. The cholesterol values obtained in the present study are quite similar to those reported in Ross 308 broiler chickens by Tavaniello et al. [38] and lower (approximately by 13.7%) compared with those observed by Kuźniacka et al. [24]. These differences between our study and cited works could be caused by the different genotypes of chickens as well as the rearing conditions and the method of feed preparation, including different seeds.

It is known that the muscle FA profile reflects both endogenous biosynthesis and diet composition. Genotype is also a source of variation; it was demonstrated that slow-growing chickens have a higher desaturation activity which allow them to efficiently convert the dietary FA precursor of the n-3 family (linolenic acid) into long-chain PUFAs such as eicosapentaenoic (C20:5, EPA) and docosahexaenoic (C22:6, DHA) acids [39]. Data in Table 4 indicated that the use of different sources of protein in the diet of broiler chickens significantly influenced the FA profile of breast muscle. Although pea replacement had no effect on total SFA (*p* > 0.05), a slight increase was observed in myristic (C14:0, *p* = 0.051) and palmitic (C16:0, *p* = 0.067) acids as compared to the SOY group. Laudadio and Tufarelli [30] found a higher level of palmitic acid and of total SFA in breast and drumstick muscles of Hubbard chickens fed with micronized–dehulled peas as compared with the soy group. Dotas et al. [23] reported higher levels of total SFA in breast meat from chickens (Ross 308) fed with different field pea inclusion levels than in meat from the soy-fed chickens. Differently, Kiczorowska et al. [25] observed that the substitution of soybean meal with micronized peas contributed to the reduction in the content of SFA in breast muscle of Ross 308 broiler chickens. An increase in the levels of palmitoleic acid (16:1n-7, *p* < 0.05) and oleic acid (18:1n-9, *p* = 0.066) was observed in the pea group. These differences may be due to a different activity of Δ-9 desaturase (stearoyl-CoA desaturase) since it has been reported that Δ-9 desaturase activity could be influenced by different factors including diet [40]. As a consequence, the total content of MUFA was higher (*p* < 0.05) in the PEA group compared to the soy group. Similarly, Dotas et al. [23] found an increase of oleic acid and total MUFA with increasing field pea inclusion levels. On the contrary, Laudadio and Tufarelli [30] found a decrease of oleic acid in breast and drumstick muscles of chickens fed peas. Regarding PUFA, the content of the precursor of the n-6 family (linoleic acid, LA), the eicosadienoic acid (C20:2 n-6) and the n-3 family precursor (linolenic acid, ALA) were significantly higher in the soy group as compared to the pea one; hence, total PUFA, n-3 and n-6 PUFA contents were also higher (*p* < 0.01) in the soy group. Similarly, Kiczorowska et al. [25] found a trend toward a lower amount of LA (*p* = 0.089) in the breast muscle of chickens fed with mixes containing pea compared to the soy group, while Dotas et al. [23] found an increased level of LA and total PUFA in breast and drumstick muscles with increasing field pea inclusion levels, in agreement with the results of Laudadio and Tufarelli [30]. The observed different effects of pea administration on FA profile could be due to the different kind of pea seeds used (micronized–dehulled pea, irradiated pea seeds or raw pea seeds), the levels of inclusion in the diet, the broiler chicken genotype and also the rearing conditions. An interaction effect of diet and genotype (*p* = 0.064) on the eicosadienoic acid (C20:2 n-6) content was observed. This result highlights that the pea inclusion lowered the levels of C20:2 in both genotypes, however the reduction was more marked among KB birds (KB-SOY: 0.45%; NR-SOY: 0.34%; KB-PEA: 0.34%; NR-PEA: 0.32%; *p* = 0.064).

In the present study, the genotype slightly affected the FA profile of breast muscle. Meat from the KB group showed a higher content of ALA (*p* < 0.05) and of eicosadienoic acid (C20:2 n-6) (*p* < 0.01) as compared with the NR group. The predominant FAs in breast muscle in all experimental groups were palmitic (C16:0) and stearic (18:0) acids such as SFA, oleic acids (C18:1n-9) such as MUFA, and linoleic (C18:2 n-6, LA) and arachidonic (20:4n-6, ARA) acids such as PUFA. As reported in a recent review [37], medium-growing and even more slow-growing chickens showed: i) a greater expression of fatty acid desaturase 1 and 2 (FADS1 and FADS2, involved in the metabolism of long-chain PUFA, LC-PUFA); ii) a higher activity of delta-6 and delta-5 desaturases which introduce double bonds in essential FA to obtain LC-PUFA; consequently, a higher LC-PUFA in the breast meat compared to fast-growing chickens. It has also been reported that there are many factors affecting PUFA metabolism and the substrate preference of the delta-6 desaturase for LA or ALA, such as species, genotype, age, sex, diet, and rearing system (reviewed in Cartoni Mancinelli et al. [41]).

The nutritional ratios were significantly affected by diet. The pea diet increased (*p* < 0.05) the n-6/n-3 ratio in broiler meat (from 16.43 to 18.92); these values are particularly high considering the current nutritional recommendations for human diets, which suggest that this ratio should not exceed 4.0. On the contrary, Laudadio and Tufarelli [30] found a significant decrease of this ratio in pea group as compared to soy group (8.57 vs. 9.88, respectively). The P/S ratio was higher (*p* < 0.01) in pea-fed birds as compared with soy-fed ones. However, the P/S values observed in the present study are favourably high (ranging from 0.78 to 0.89). From a nutritional point of view, a higher P/S ratio is recommended; indeed, it should be increased to above 0.4 [42]. The AI and TI represent criteria for evaluating the level and interrelation through which some FA may have atherogenic or thrombogenic properties, respectively. Diet and genotype did not affect the AI and TI indices (*p* > 0.05). Similarly, Laudadio and Tufarelli [30] did not find any significant effect of pea inclusion on the above-mentioned indices with similar values for AI (0.55 and 0.58 for soy and pea, respectively), but lower values for TI (0.55 and 0.56, respectively) as compared to our results. The AI and TI values found in the current study can be considered similar with those reported by Banaszak et al. [43] in ducks fed yellow lupin.

## 4. Conclusions

The total replacement of soybean with raw field peas as the main protein and energy source in diets of broiler chickens did not show adverse effects on carcass characteristics and physico-chemical properties of chicken meat. However, it has a marked effect on fatty acid composition. A pea diet determined a significant increase in the total content of MUFA and a decrease in the total PUFA and in n-3 and n-6 PUFA contents; hence, breast meat from the pea group showed lower P/S and higher n-6/n-3 ratios as compared to the soy group, with negative effects on the nutritional value of intramuscular fat. Chicken genotype significantly affected slaughter traits: New Red chickens showed a greater productive capacity. Fatty acid composition was slightly affected by genotype. Further studies are needed to find the better inclusion rate in order to promote the use of peas in broiler diets. In other words, to find the right compromise between environmental and economic sustainability of the diet and promotion of the nutritional quality of the meat.

## Figures and Tables

**Table 1 animals-12-02849-t001:** Ingredients, chemical composition, energy content and fatty acid composition of the diets.

Feed Administered ^a^	SOY	PEA
Ingredients, g/kg DM		
Wheat bran	491.5	474.6
Durum wheat	254.2	262.7
Corn meal	152.6	127.1
Fava bean	67.8	67.8
Pea bean	-	67.8
Soybean flaked, 37% CP	33.9	-
Total	1000	1000
Analyzed results		
DM, g/100 g	88.9	89.2
CP, g/100 g DM	18.6	18.6
EE, g/100 g DM	4.82	4.45
Ash, g/100 g DM	4.97	4.83
Calculated analysis		
CF, g/100 g DM	8.16	8.14
Lys, g/100 g DM	0.67	0.69
Met, g/100 g DM	0.25	0.24
aME, MJ/100 g DM	1.33	1.32
Fatty acids, % of total fatty acids		
C14:0	0.10	0.09
C16:0	15.87	14.60
C18:0	1.45	2.33
ƩSFA	17.41	17.03
C16:1	0.06	0.08
C18:1n9	20.91	21.34
ƩMUFA	20.96	21.42
C18:2 n-6	57.50	56.18
C18:3 n-3	4.12	5.37
ƩPUFA	61.63	61.55
n6/n3	13.95	10.46

^a^ SOY = control diet including flaked soybean (3.39 g/100 g DM); PEA = experimental diet including pea beans (6.78 g/100 g DM); DM = dry matter; CP = crude protein; EE = ether extract; CF = crude fibre; Lys = lysine; Met = methionine; aME = apparent metabolizable energy; SFA = saturated fatty acids; MUFA = monounsaturated fatty acids; PUFA = polyunsaturated fatty acids; n6/n3 = n6 PUFA/n3 PUFA.

**Table 2 animals-12-02849-t002:** Effect of genotype and diet on carcass traits of broiler chickens.

	Diet (D) ^a^		Genotype (G) ^b^		SEM	Significance		
	SOY	PEA	KB	NR		D	G	D × G
Final body weight, g	2800.0	2677.5	2560.0	2917.5	44.98	ns	**	ns
Carcass weight, g	1709.0	1641.5	1540.0	1810.5	28.58	ns	**	ns
Carcass yield, %	61.02	61.18	60.12	62.08	0.28	ns	**	ns
Breast weight, g	372.4	366.0	327.9	410.5	8.11	ns	**	*
Breast yield, %	21.78	22.07	21.18	22.67	0.23	ns	**	*
Legs weight, g	560.8	538.3	498.5	600.6	9.78	ns	**	ns
Legs yield, %	32.80	32.80	32.38	33.22	0.25	ns	ns	ns
Wings weight, g	175.7	169.0	160.9	183.8	2.91	ns	**	ns
Wings yield, %	10.31	10.36	10.50	10.17	0.12	ns	ns	ns

^a^ SOY = control diet including flaked soybean (3.39 g/100 g DM); PEA = experimental diet including pea beans (6.78 g/100 g DM); ^b^ KB = Kabir Rosso Plus; NR = New Red. Legs = Thigh and drumstick; SEM = standard error mean; ns = not significant (*p* > 0.05); * *p* < 0.05; ** *p* < 0.01.

**Table 3 animals-12-02849-t003:** Effect of genotype and diet on the physico-chemical properties of the breast muscle of broiler chickens.

	Diet (D) ^a^		Genotype (G) ^b^		SEM	Significance		
	SOY	PEA	KB	NR		D	G	D × G
pH_24_	5.84	5.84	5.80	5.88	0.02	ns	*	ns
Color 24 h								
*L**	53.98	53.91	54.16	53.73	0.51	ns	ns	ns
*a**	3.22	3.19	3.52	2.89	0.17	ns	ns	ns
*b**	5.81	5.54	5.49	5.85	0.39	ns	ns	ns
WHC, %	13.90	13.78	14.28	13.39	0.26	ns	ns	ns

^a^ SOY = control diet including flaked soybean (3.39 g/100 g DM); PEA = experimental diet including pea beans (6.78 g/100 g DM); ^b^ KB = Kabir Rosso Plus; NR = New Red. WHC = Water holding capacity; SEM = standard error mean; ns = not significant (*p* > 0.05); * *p* < 0.05.

**Table 4 animals-12-02849-t004:** Effect of genotype and diet on cholesterol and total lipid content, fatty acid composition and nutritional indices in the breast muscle of broiler chickens.

	Diet (D) ^a^		Genotype (G) ^b^		SEM	Significance		
	SOY	PEA	KB	NR		D	G	D × G
Cholesterol, mg/100 g	39.25	42.99	42.47	39.96	1.30	ns	ns	ns
Lipids, %	1.16	1.16	1.21	1.12	0.04	ns	ns	ns
Fatty Acids ^b^, % of total fatty acids								
C12:0	0.06	0.08	0.08	0.06	0.01	ns	ns	ns
C14:0	0.42	0.49	0.45	0.46	0.02	ns	ns	ns
C16:0	26.50	27.87	27.06	27.31	0.36	ns	ns	ns
C16:1	1.45	1.98	1.54	1.89	0.11	*	ns	ns
C17:0	0.21	0.20	0.21	0.20	0.01	ns	ns	ns
C17:1	0.09	0.09	0.09	0.09	0.01	ns	ns	ns
C18:0	11.57	11.34	11.65	11.26	0.19	ns	ns	ns
C18:1n9	25.41	26.95	26.54	25.82	0.41	ns	ns	ns
C18:2 n-6	24.80	22.74	23.96	23.59	0.27	**	ns	ns
C18:3 n-6	0.11	0.10	0.11	0.09	0.01	ns	ns	ns
C18:3 n-3	1.00	0.78	0.97	0.81	0.03	**	*	ns
C20:2n6	0.39	0.33	0.40	0.33	0.01	*	**	ns
C20:3n6	0.40	0.42	0.41	0.41	0.02	ns	ns	ns
C20:4 n-6	5.41	4.78	4.69	5.50	0.26	ns	ns	ns
C20:5 n-3	0.08	0.06	0.06	0.08	0.01	ns	ns	ns
C22:4n-6	1.15	1.03	1.02	1.16	0.06	ns	ns	ns
C22:5n3	0.58	0.44	0.46	0.56	0.04	ns	ns	ns
C22:6 n-3	0.37	0.31	0.31	0.37	0.03	ns	ns	ns
ƩSFA	38.76	39.97	39.44	39.29	0.34	ns	ns	ns
ƩMUFA	26.95	29.02	28.17	27.81	0.46	*	ns	ns
ƩPUFA	34.29	31.00	32.39	32.90	0.54	**	ns	ns
Ʃn-6	32.26	29.40	30.58	31.08	0.49	**	ns	ns
Ʃn-3	2.03	1.60	1.80	1.82	0.06	**	ns	ns
Nutritional indices ^c^								
n-6/n-3	16.43	18.92	17.48	17.87	0.47	*	ns	ns
P/S	0.89	0.78	0.83	0.84	0.02	**	ns	ns
AI	0.46	0.50	0.48	0.48	0.02	ns	ns	ns
TI	1.08	1.17	1.13	1.12	0.03	ns	ns	ns

^a^ SOY = control diet including flaked soybean (3.39 g/100 g DM); PEA = experimental diet including pea beans (6.78 g/100 g DM); ^b^ KB = Kabir Rosso Plus; NR = New Red. ^b^ SFA = saturated fatty acids; MUFA = monounsaturated fatty acids; PUFA = polyunsaturated fatty acids. ^c^ P/S = PUFA/SFA ratio; AI = atherogenic index; TI = thrombogenic index. SEM = standard error mean. ns = not significant (*p* > 0.05); * *p* < 0.05; ** *p* < 0.01.

## Data Availability

Data presented in this study are available on request from corresponding author.

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
