# Peer review of "Carcass and Meat Quality Traits of Medium-Growing Broiler Chickens Fed Soybean or Pea Bean and Raised under Semi-Intensive Conditions"

_animals, 2022, doi:10.3390/ani12202849_

Round 1

Reviewer 1 Report

Dear Authors,

I have revised the manuscript entitled "Soybean vs. Pea Bean in the Diet of Medium-Growing Broiler Chickens Raised under Semi-Intensive Conditions of Inner Mediterranean Areas: Carcass and Meat Quality Traits" (identified as animals-1957672). The focus of the manuscript is to study the effects of the total replacement of flaked soybean with peas on carcass characteristics and broiler meat quality. Each diet was tested on two different commercial hybrids, both slow growing. As a whole, the manuscript is well structured and reflects the journal's standards in all its parts. The results are well summarized and well discussed, and the conclusions are adequate. However, I have some concerns about the introduction, which, in my opinion, does not adequately focus on some relevant aspects. In fact, as stated by the authors, the replacement of soy products with legume grains has been extensively addressed. Among them, the use of peas has also already been evaluated in multiple research studies. So, I believe that the authors need to better argue their choices, emphasizing the regions based on which pea and no other legumes were tested (this is not evident little or at all in the introduction), as well as the reasons justifying these choices in relation to the agricultural and social context (i.e., "Semi-Intensive Conditions of Inner Mediterranean Areas). In the title, this was rightly stated but, conversely, not at all is recalled later. As far as I know, pea cultivation in the inner areas of the southern Apennines is practiced to alleviate the problems of cereal monoculture, as well as to improve the productive performance of mixed agro-forestry systems and connect these with livestock farming in an eco-sustainable farming perspective (as reference I suggest https://doi.org/10.1007/s10457-018-0316-5). In addition, unlike other legumes, the pea is a flexible crop, fitting into many crop rotations due to its potential winter sowing and rapid seed ripening, which allows them to escape summer drought and high temperatures (as references I suggest https://doi.org/10.2134/agronj2008.0085 and https://doi.org/10.1017/S0021859607007289). These aspects, among others, may constitute valid motivations for the use of pea in the Mediterranean environment, thus justifying its use in feedstuffs on a local scale and for alternative livestock farms, such as the one described in the study. An additional aspect that the authors are invited to consider is the description of the allocation of dietary treatments among experimental groups, which in my opinion is unclear in the abstract (lines 25 and 26) and, partially in the materials and methods (L 102). Finally, the authors are asked to check for typos along the text carefully (e.g., "final1" in line 17, the absence of the period, spacings, etc.). Hoping to have contributed to improving the manuscript quality. Good works.

Author Response

Thanks to the comments and suggestions the resulting manuscript is a great improvement over the text that was originally submitted.

Reviewer 2 Report

The article is well organized, but some of the information in the different sections could be improved.

Major comments:

The Materials and Methods section, L94-102, L105-109, and Table 1 (except fatty acids) are exactly the same as Fatica et al. (2022). An earlier study indicated the content and chemical composition of the ration: therefore, it should be cited. A number of results were cited, such as those for body weight, feed intake, and feed conversion ratio for L167-168. On the other hand, it is not recommended that the sentences used in another study in the material method section are used one-to-one. This may lead to misunderstandings.

In Table 2, only the final body weight is given as a growth parameter. To reach an accurate evaluation of growth performance, it would be more accurate to incorporate the other performance data gathered by the authors in the previous study.

L170, L207, L231: Despite a genotype effect in Table 2, Table 3, and Table 4, lettering is not included. There is a similar situation in Figure 1 since the DxG interaction is significant (L203). Furthermore there is a similar situation in Table 4 for diet.

Since diet and genotype interact insignificantly, making an inference stated in L284-286 may be incorrect.

Minor comments for “Introduction” section

It is advisable to divide the introduction into a few paragraphs

L54: “august” should be corrected as “August”

L59:  Pisum sativum” and “Vicia faba” should be corrected as “Pisum sativum L.” and “Vicia faba L.

L64: “sources” instedad of “components”

L78-80: a few references should be given

Minor comments for “Materials and Methods” section

L94-95: Country should be stated

L127-128, L207 (Table 3): “L*”, “a*, ”b*” should be italicized

L130, L137, L139, L140: There should be space between words “number” and “degree”

L142: “Component” instedad of “components”

Minor comments for “Results and Discussion” section

L165: “Discussion” instedad of “discussion”

L171, L207, L231: A brief explanation of the abbreviations (SOY and PEA) should also be placed beneath the table.

L176, L186: “BW” instedad of “body weight”

L203: “Breast weight (a) and breast yield (b)”instedad of “(a) Breast weight and (b) breast yield”

L203: There should be space between words “±” and “SD”

L218: “WHC” instedad of “water holding capacity”

L220: the Word “any” should be added between “indicate” and “problems”.

L240: “Hubbard JA757” instedad of “JA757”

L256-257, L265, L276, L283: "p = 0.051", "p = 0.067" , “p = 0.066”, p = 0.089”, p = 0.066” should be removed or a p-value should be given in all tables and figures.

Author Response

(The authors gave the same response as above.)

Round 2

Reviewer 1 Report

Dear Author(s),
I have reviewed the recent version of your manuscript identified as "animals-1957672". I find that the additions made by the authors have improved the manuscript, allowing it to be diversified, at least as context analysis, from other studies, focused on developing alternative schemes to soy in poultry feeding. Therefore, I have no qualms in suggesting the publication of the revised version of the manuscript. I offer my congratulations and wish you good luck. Regards